# Unlocking the potentials of cyanobacterial photosynthesis for directly converting carbon dioxide into glucose

Shanshan Zhang [1,2,3,4,7], Jiahui Sun[1,2,3,4,7], Dandan Feng[1,2,3], Huili Sun[1,2,3,4], Jinyu Cui[1,2,3], Xuexia Zeng[1,2,3], Yannan Wu[1,2,3], Guodong Luan [1,2,3,4,5] & Xuefeng Lu [1,2,3,4,5,6]

Glucose is the most abundant monosaccharide, serving as an essential energy source for cells in all domains of life and as an important feedstock for the biorefinery industry. The plant-biomass-sugar route dominates the current glucose supply, while the direct conversion of carbon dioxide into glucose through photosynthesis is not well studied. Here, we show that the potential of *Synechococcus elongatus* PCC 7942 for photosynthetic glucose production can be unlocked by preventing native glucokinase activity. Knocking out two glucokinase genes causes intracellular accumulation of glucose and promotes the formation of a spontaneous mutation in the genome, which eventually leads to glucose secretion. Without heterologous catalysis or transportation genes, glucokinase deficiency and spontaneous genomic mutation lead to a glucose secretion of 1.5 g/L, which is further increased to 5 g/L through metabolic and cultivation engineering. These findings underline the cyanobacterial metabolism plasticities and demonstrate their applications for supporting the direct photosynthetic production of glucose.

Glucose is the most abundant monosaccharide molecule in nature. The breakdown of glucose provides energy and carbon materials in cells throughout all domains of life, powering the cellular machinery by diverse glycolytic pathways, including the Embden–Meyerhof–Parnas (EMP) pathway, the oxidative pentose phosphate (OPP) pathway, and the Entner–Doudoroff (ED) pathway[1,2]. As monomers, glucose, and its derivatives are also involved in the synthesis of various macromolecules and cellular components[3,4]. Moreover, glucose also serves as an important feedstock in the biorefinery industry, supporting the cultivation of multiple microbial cell factories for green biomanufacturing of fuels, chemicals, and pharmaceuticals[5–7]. In nature, glucose is mainly synthesized through the photosynthesis of plants and algae and exists as monomers of polysaccharides in plant/algae biomass, e.g., cellulose and starch. The plant-biomass-sugar route dominates the current massive glucose supply, whose economic feasibility is influenced by multiple parameters, such as plant-cultivation cycles, biomass-collection radius, and pre-treatment costs[8–11].

Against the backdrop of the global climate crisis and worsening food shortages, developing more efficient, continual, and industrial glucose production routes would be valuable[12,13]. In recent years, direct conversion of carbon dioxide into glucose, glucose precursors, and glucose polymers has been achieved by chemical-biochemical, electrochemical-biological, and in vitro cascade enzymatic routes[14–16]. In contrast, continuous glucose production has not been successfully

[1]Key Laboratory of Biofuels, Qingdao Institute of Bioenergy and Bioprocess Technology, Chinese Academy of Sciences, No. 189 Songling Road, Qingdao, Shandong 266101, China. [2]Shandong Energy Institute, No. 189 Songling Road, Qingdao, Shandong 266101, China. [3]Qingdao New Energy Shandong Laboratory, Qingdao, Shandong 266101, China. [4]College of Life Science, University of Chinese Academy of Sciences, 100049 Beijing, China. [5]Dalian National Laboratory for Clean Energy, Dalian, Liaoning 116023, China. [6]Laboratory for Marine Biology and Biotechnology, Qingdao National Laboratory for Marine Science and Technology, Qingdao, Shandong 266237, China. [7]These authors contributed equally: Shanshan Zhang, Jiahui Sun. ✉e-mail: luangd@qibebt.ac.cn; lvxf@qibebt.ac.cn

linked directly to photosynthesis. In photoautotrophs, e.g., higher plants and algae, glucose is synthesized as storage for carbon and energy and plays important regulatory roles. Glucose metabolism possesses complex interactions with photosystems, disturbs the synthesis and metabolism of pigments, and might even inhibit the photosynthetic activities[17–19]; thus, free glucose is rarely synthesized or accumulated in excess in photosynthetic cellular metabolism. In cyanobacteria, a group of oxygenic prokaryotic microalgae, some progress has been made to facilitate the direct synthesis and secretion of natural or non-natural sugars through genetic manipulations[20–22]. However, the photosynthetic production of glucose has not yet been well-studied. The recombinant strains can only produce limited amounts of glucose accompanied by the production of other sugars, suggesting that more detailed mechanisms of glucose metabolism in photoautotrophs remain to be disclosed[23,24].

In this work, we aim to engineer a direct and stable conversion of carbon dioxide to glucose through cyanobacteria photosynthesis. In a model cyanobacterium *Synechococcus elongatus* PCC 7942 (hereafter PCC 7942 for short), we identify the native glucokinase activity as the bottleneck restricting the metabolism potential for glucose synthesis. Targeted knockout of two glucokinase genes disturbs the carbohydrate metabolism and activates a metabolic flux towards glucose through the sucrose metabolism network, which is generally considered as a specialized response to osmotic stress. The enhanced glucose synthesis promotes the enrichment of a specific spontaneous genomic mutation on the chromosome of PCC 7942, which facilitates efficient glucose secretion. By implementing multiple omics approaches combined with systematic genetic manipulations, we clarify the pathways and mutations leading to glucose synthesis and secretion and optimize the glucose synthesis performances of the recombinant strains. Through subsequent metabolic engineering and cultivation optimization, the glucose secreted by the engineered strain surpasses 5 g/L during long-term cultivation, accounting for up to 70% of the fixed carbon source.

## Results
### Blocking glucose consumption by knocking out the glucokinases

There are usually three steps to achieve efficient microbial production of the target metabolites: to construct pathways for the synthesis of the products[25,26], to eliminate reactions that consume such

products[27,28], and to facilitate the mechanisms for product secretion[20,23]. Previous studies have shown that PCC 7942 can uptake extracellular glucose for biomass production by using heterologous glucose transporters, demonstrating their natural capacities for intracellular glucose consumption[29]. Two glucokinase genes (*Synpcc7942_0221*, *glk1*; *Synpcc7942_2111*, *glk2*) that phosphorylate glucose to initiate the EMP or OPP pathway have been annotated on the PCC 7942 genome, so they were selected as knockout targets to block intracellular glucose re-assimilation.

The manipulations were performed in parallel using a wild-type PCC 7942 (WT) and a recombinant strain carrying an *E. coli* sourced glucose transporter GalP (SZ14, Supplementary Figs. 1 and 2). Previously, GalP has proven to be effective for facilitating glucose absorption in PCC 7942[29,30], and here we use it to evaluate the impacts of glucokinase deficiency on the glucose consumption capacities of PCC 7942. Due to the polyploidy characteristics of cyanobacteria, passages with antibiotics-selection were necessary to isolate the desired homozygotes. Recombinant strains carrying an inactivated glucokinase gene (*glk1* or *glk2*) were easily obtained, but homozygote with simultaneously inactivated *glk1* and *glk2* in WT was difficult to isolate. Among several independent attempts, only one process resulted in a homozygote strain (SZ3) with a tailored genotype (completely blocked *glk1* and *glk2*) after six weeks of continuous passages, while all the others failed (meaning that the wild-type copy of *glk1* or *glk2* could not be removed). The results indicated that glucokinase might have essential physiological functions, and potential secondary genomic mutations might be necessary for cellular acclimation to the deficiency. In the SZ14 strain carrying the GalP transporter, the homozygote mutant (SZ17) with simultaneously blocked *glk1* and *glk2* was more easily isolated, which indicates that the GalP activity might buffer the disturbance caused by the glucokinase deficiency.

Enzymatic assays revealed that both genes (*glk1* and *glk2*) contributed to the glucokinase activity in PCC 7942, and their simultaneous knockout reduced glucokinase activity by about 70% in SZ3; as for the SZ14 strain with GalP transporter, glucokinase activity was even more significantly reduced by knocking out the two genes (in SZ17) to a lower level than that of SZ3 (Fig. 1a). The residual glucokinase activities could be attributed to other non-specific kinases (e.g., fructokinase and phosphofructokinase). The higher glucokinase activity residual in SZ3 than that in SZ17 indicated that the wild-type background seemed essential to maintain a certain degree of glucokinase

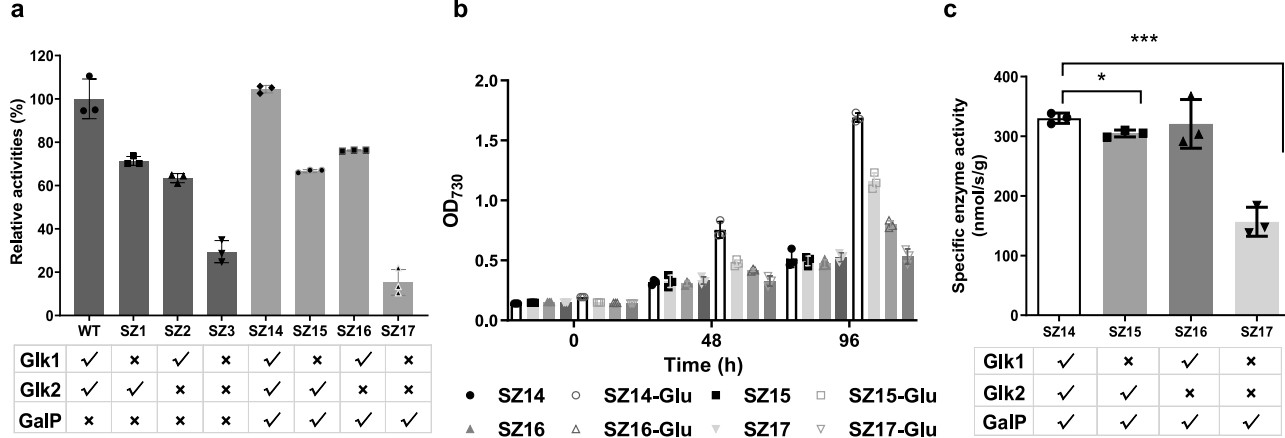

**Fig. 1 | Effects of glucokinase deficiency on PCC 7942 with (SZ14) and without (SZ3) heterologous glucose transporter. a** Relative glucokinase activity of WT, SZ1 (Δ*glk1*), SZ2 (Δ*glk2*), SZ3 (Δ*glk1*-Δ*glk2*), SZ14 (Δ*NS3::galP*), SZ15 (Δ*NS3::galP*-Δ*glk1*), SZ16 (Δ*NS3::galP*-Δ*glk2*), and SZ17 (Δ*NS3::galP*-Δ*glk1*-Δ*glk2*) grown under autotrophic conditions. **b** Growth of SZ14, SZ15, SZ16, and SZ17 in BG11 culture medium with or without supplemented glucose. **c** Glucokinase activity of SZ14,

SZ15, SZ16, and SZ17 grown under mixotrophic conditions supplemented with 30 mM glucose. Statistical analysis was performed using two-tailed unpaired Student's *t* test (*$p < 0.05$, ***$p < 0.001$). The *p* values in (c) are (from left to right) 0.013, 0.0003. Data are presented as mean values ± SD (*n* = 3 biological replicates). Source data are provided as a Source Data file.

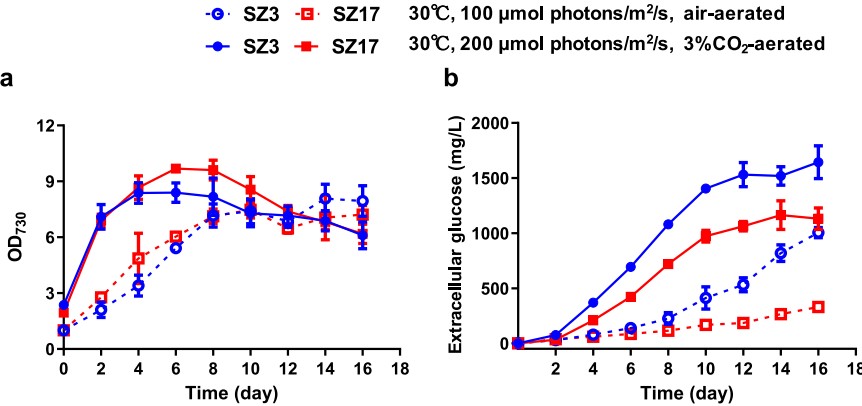

**Fig. 2 | Cell growth and glucose secretions of the glucokinase-deficient strains.** Cell densities (**a**) and extracellular glucose concentratons (**b**) of the the glucokinasedeficient strains SZ3 and SZ17 were measured and compared. SZ3 and SZ17 were cultivated under two conditions with different illumination strengths and carbon supply. Empty symbols and dotted lines represent the profiles from

cultivation conditions of 30 °C, 100 μmol photons/m²/s and air-aerated; and solid symbols and solid lines represent profiles from the cultivation conditions of 30 °C, 200 μmol photons/m²/s and 3% $CO_2$-aerated. Data are presented as mean values ± SD (*n* = 3 biological replicates). Source data are provided as a Source Data file.

activity, so the recombinant strain underwent long-term domestication to adapt to the loss of the activities by uncertain mechanisms.

The mixotrophic cultivation results confirmed that intracellular glucose consumption was prevented in PCC 7942 as per the design. Supplementation of glucose (30 mM) improved the biomass production of the SZ14 strain (carrying the GalP transporter) by three folds, whereas no growth improvement was detected in the glucokinase-deficient strain, which indicated a loss in glucose utilization capacity (Fig. 1b). In addition, the supplementation of glucose led to the recovery of glucokinase activity in the deficient strains (Fig. 1c), indicating an adaptive response of the PCC 7942 metabolism toward exogenous glucose including stimulation of non-specific glucokinase activities.

## Glucokinase deficiency resulted in glucose secretion

The enzymatic assay results indicated that glucokinase activities might be essential for PCC 7942 cells to maintain metabolism homeostasis, while the dependence could be partially relieved by introducing the glucose transporter GalP. Given the effects of GalP on native glucokinase dependence, we hypothesized that the glucokinase deficiency in PCC 7942 would lead to an abnormal accumulation of specific metabolites, that could be exported by GalP to relieve the potential toxicity or feedback stress. To test this hypothesis, we investigated the secreted metabolite profiles of the glucokinase-deficient strains (Supplementary Fig. 3a) and found that significant amounts of glucose were accumulated in the culture medium. Under the adopted cultivation conditions (BG11 medium, 100 μmol photons/m²/s, air bubbling for carbon supply), 1.0 g/L and 0.33 g/L glucose were synthesized by SZ3 (WT-Δglk1-Δglk2) and SZ17 (WT-Δglk1-Δglk2-ΔNS3::galP) strain respectively (Fig. 2). In addition, the accumulation of glucose occurred in the same pace as the cellular biomass production, which indicates that the extracellular glucose was synthesized and secreted by the living cells rather than from the contents of lysed cells. When carbon supply and illumination were further enhanced (3% $CO_2$ v/v in air, 200 μmol photons/m²/s), the glucose production of SZ3 and SZ17 increased to 1.64 g/L and 1.16 g/L, respectively; and the rates of the two strains for biomass accumulations were also improved. During the process, intracellular glucose concentrations in the glucokinase deficiency strains also increased (Supplementary Fig. 3b). As for SZ3, about 5.6 mg/L/OD730 glucose (over 70-fold higher than that of the control) was accumulated inside the cells on Day 6, and for SZ17, the intracellular glucose concentration reached up to 2.8 mg/L/OD730.

To exclude the potential contributions of organic carbon source in the BG11 medium to glucose synthesis, we removed the citrate and

ferric amine citrate components from the medium, but found no negative effect on glucose production, although cell growth of the recombinant strains was slightly diminished (Supplementary Fig. 4a–c). We also performed isotope labeling cultivation to evaluate the contributions from organic carbon sources. As shown in Supplementary Fig. 4d–i, when the recombinant *Synechococcus* cells were cultivated with ¹³C labeled $NaHCO_3$, the synthesized glucose could be rapidly labeled by ¹³C, and the addition or removal of citrate and ferric amine citrate caused minor impacts on the labeling rates. Thus, we could propose that the glucose secreted by recombinant strains was dominantly from the conversion of inorganic carbon dioxide. Compared with previous reports, photosynthetic production of glucose by cyanobacteria was improved in this work. Over tenfold higher glucose productivity (0.27 g/L/OD730 versus 0.027 g/L/OD730) was achieved without the introduction or overexpression of any catalytic enzymes in *Synechococcus*. Moreover, the glucose synthesis of recombinant PCC 7942 continued during the rapid growth phase independent of any environmental stress inductions (e.g., salts stress or dark treatment)[23,24]. In addition, the SZ3 strain secreted most of the glucose (>95%, approximately 0.27 g/L/OD730 versus 5.6 mg/L/OD730) independently of heterologous transporters, suggesting that unknown glucose transportation mechanisms were activated by the glucokinase deficiency.

## Glucose synthesis is dominated by the sucrose metabolism pathway

Glucose is generally not recognized as a main intracellular metabolite in PCC 7942, so we did not expect that blocking the glucose consumption capacity would shift a large portion of the carbon flux toward glucose synthesis. Theoretically, glucose can be generated through dephosphorylation of glucose-1-phosphate or glucose-6-phosphate catalyzed by glucose-1-phosphatase (EC 3.1.3.10) or glucose-6-phosphatase (EC 3.1.3.9). Although these genes have not been identified in PCC 7942, other phosphatases (alkaline phosphatase, *Synpcc7942_1392*; inorganic diphosphatase, *Synpcc7942_1383*; fructose-1,6-bisphosphatase II/sedoheptulose-1,7-bisphosphatase, *Synpcc7942_0505*; fructose-1,6-bisphosphatase, *Synpcc7942_2335*; myo-inositol-monophosphatase, *Synpcc7942_2582*) might catalyze the reactions using glucose-1-phosphate or glucose-6-phosphate as non-specific substrates. Considering the amount of accumulated glucose, a more possible route was the sucrose hydrolyzation pathway, through which UDP-glucose and fructose-6-phosphate would be converted into sucrose catalyzed by sucrose-6-phosphate synthase (*sps*, *Synpcc7942_0808*), and the generated sucrose could be hydrolyzed into glucose and fructose (*invA*, *Synpcc7942_0397*) (Fig. 3a). To

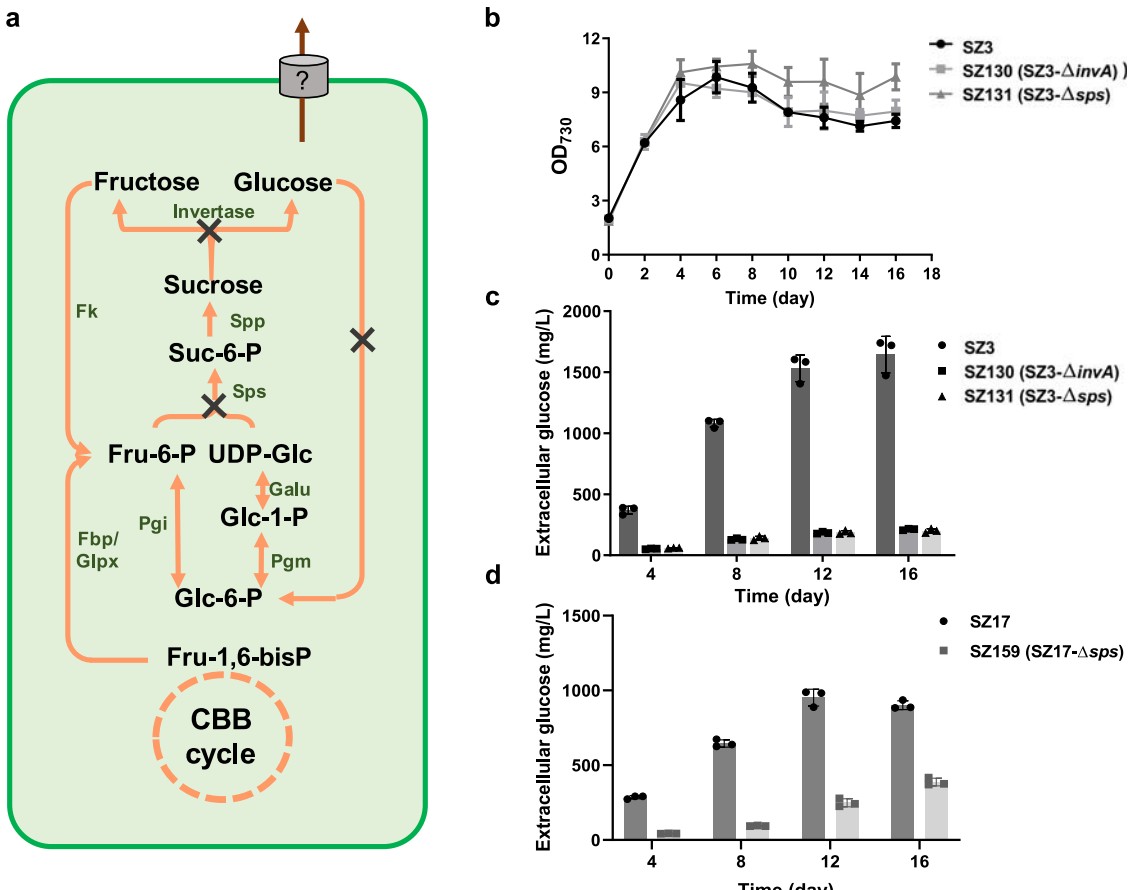

**Fig. 3 | Contribution of sucrose metabolism pathway to the glucose synthesis of the SZ3 and SZ17 strains. a** Schematic metabolic network of glucose generation in PCC 7942. Cell growth (**b**) and glucose production (**c**) of SZ3, SZ130 (SZ3-*ΔinvA*), and SZ131 (SZ3-*Δsps*). **d** Glucose production of SZ17 and SZ159 (SZ17-*Δsps*). Cultivation was conducted at 30 °C, 200 μmol photons/m$^2$/s, and bubbled with 3% $CO_2$. Data are presented as mean values ± SD ($n$ = 3 biological replicates). CBB cycle Calvin–Benson–Bassham cycle, Fru-1,6-bisP fructose-1,6-bisphosphate, Fru-6-P fructose-6-phosphate, UDP-Glc UDP-glucose, Suc-6-P sucrose-6-phosphate, Glc-6-P glucose-6-phosphate, Glc-1-P glucose-1-phosphate, Sps sucrose-phosphate synthase, Spp sucrose-phosphate phosphatase, Fk fructokinase, Pgi glucose-6-phosphate isomerase, Pgm phosphoglucomutase, Fbp fructose-1,6-bisphosphatase I, Glpx fructose-1,6-bisphosphatase II/sedoheptulose-1,7-bisphosphatase, Galu UTP-glucose-1-phosphate uridylyltransferase. Source data are provided as a Source Data file.

confirm this hypothesis, we knocked out the two genes (SZ130, SZ3-*ΔinvA*; SZ131, SZ3-*Δsps*; Supplementary Figs. 5 and 6), and found that the growth rate was not affected while glucose secretion was reduced by about 90% to 0.2 g/L (Fig. 3b, c). The same phenomenon was observed for the SZ17 strain (Fig. 3d).

The results indicated that the sucrose synthesis pathway dominated the glucose accumulation in the glucokinase-deficient strains, which was counterintuitive because sucrose synthesis is usually recognized as an inducible physiological protective mechanism to resist hypersaline stress[31]. To investigate if a significant metabolic flux was maintained through the sucrose synthesis pathway in the glucokinase-deficient strains independent of hypersaline stress induction, we constructed an SZ130 derived strain, namely SZ153, carrying a heterologous sucrose transporter sucrose-proton permease (CscB) (Supplementary Figs. 7a, b), in which the *invA* deficiency would block the potential sucrose hydrolysation, and the isopropyl-beta-D-thiogalactopyranoside (IPTG) induced *cscB* expression could facilitate the secretion of accumulated sucrose (Fig. 4a). As shown in Fig. 4b, under standard conditions, there was no significant difference between the extracellular glucose produced by the two strains, whereas sucrose was secreted continuously only in SZ153 (to 350 mg/L in 8 d) with IPTG induced *cscB* expression. By expressing the CscB, the intracellular concentration of sucrose in SZ153 was also decreased from 10 mg/L/OD$_{730}$ to about 2 mg/L/OD$_{730}$, while the intracellular glucose concentration was not influenced (Supplementary Figs. 7c, d).

Thus, it could be confirmed that a sucrose synthesis flux was maintained in PCC 7942 cells regardless of hypersaline stress.

Further experiments revealed that the active sucrose synthesis in PCC 7942 was not related to glucokinase deficiency. The sucrose secretion through CscB under non-hypersaline conditions was also observed for the wild-type background. In the previously constructed strain FL92 (WT-*ΔNS3::P$_{lac}$-cscB*)[32], IPTG-induced *cscB* expression improved extracellular sucrose secretion. The knockout of *invA* (named SZ21, Supplementary Fig. 8) improved sucrose secretion from 16 to 80 mg/L when no IPTG was added (*cscB* expressed under a "leaky" state). When 0.1 mM IPTG was added, sucrose secretion by all three strains further increased to over 190 mg/L, and less glucose secretion was observed in the SZ21 and FL92 strains with functional glucokinases (Fig. 4c). Based on the above results, we assumed that an active sucrose metabolism cycle existed in PCC 7942 and that the deficiency of glucokinase blocked glucose reutilization and further promoted the subsequent accumulation and secretion. In the sucrose metabolism deficient strains, a certain amount of glucose was still synthesized, indicating that there were additional pathways contributing to the glucose synthesis (e.g., potential non-specific dephosphorylation activity) in PCC 7942.

**A specific SNP facilitated efficient glucose secretion**

The glucokinase deficiency resulted in intracellular accumulation of glucose, whereas the mechanisms facilitating subsequent secretion

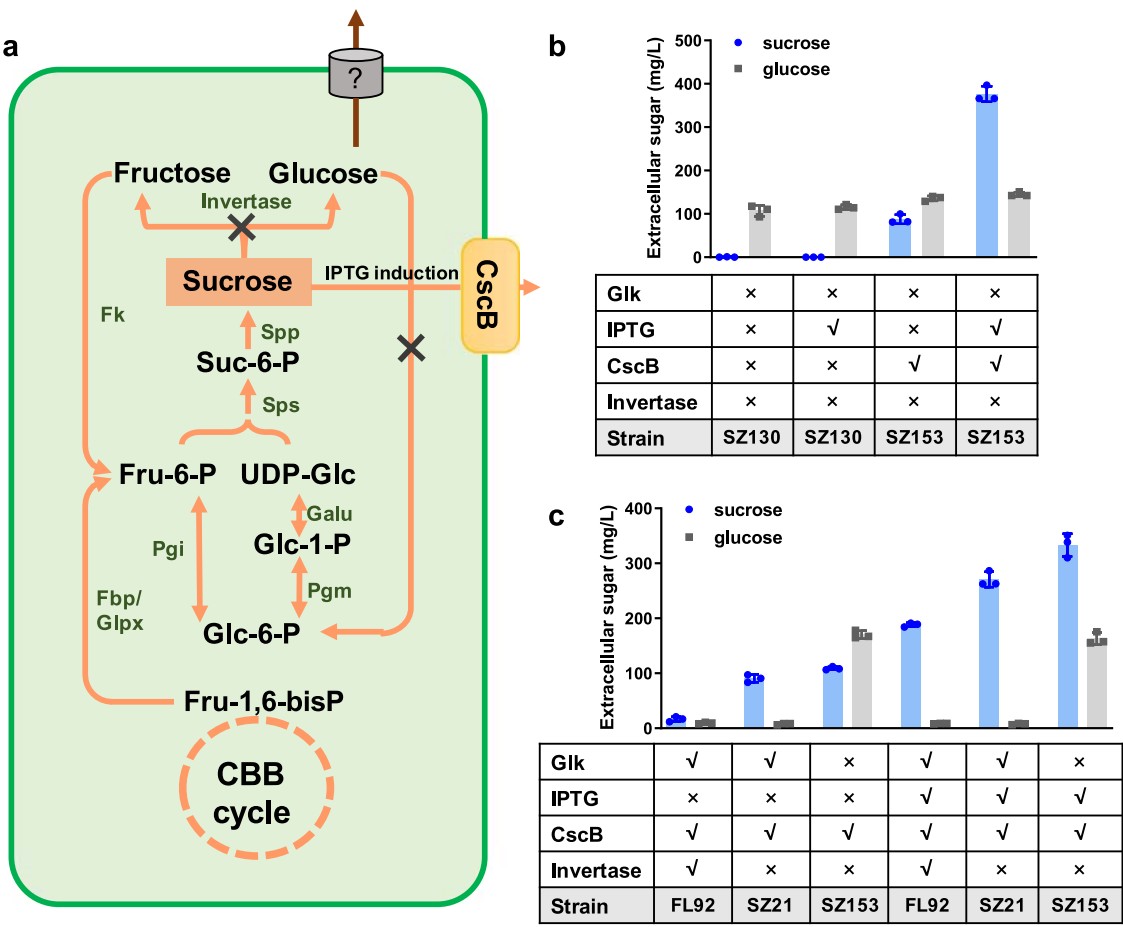

**Fig. 4 | Effects of the CscB transporter on sucrose and glucose secretion of PCC 7942 derived strains. a** Engineering strategies to facilitate sucrose secretion in PCC 7942. **b** Sucrose/glucose secretion in SZ130 (Δglk1-Δglk2-ΔinvA) and SZ153 (Δglk1-Δglk2-ΔinvA-ΔNS3::cscB) with or without IPTG induction. **c** Extracellular glucose and sucrose accumulations of FL92 (ΔNS3::cscB), SZ21 (ΔinvA-ΔNS3::cscB), and SZ153 (Δglk1-Δglk2-ΔinvA-ΔNS3::cscB) with or without IPTG induction. The cells were grown in BG11 for 8 d. Data are presented as mean values ± SD. All sample size n (biological replicates) is 6. Glk√ and Glk×: intact and deficient glucokinase, respectively; IPTG√: strain grown in the presence of 0.1 mM IPTG; IPTG×: no IPTG added to the BG11 medium; CscB×: no CscB in the strain; CscB√: strain expressing CscB; Invertase √ and Invertase×: intact and deficient invertase, respectively; CBB cycle Calvin–Benson–Bassham cycle, Fru-1,6-bisP fructose-1,6-bisphosphate, Fru-6-P fructose-6-phosphate, UDP-Glc UDP-glucose, Suc-6-P sucrose-6-phosphate, Glc-6-P glucose-6-phosphate, Glc-1-P glucose-1-phosphate, Sps sucrose-phosphate synthase, Spp sucrose-phosphate phosphatase, Fk fructokinase, CscB sucrose permease, Pgi glucose-6-phosphate isomerase, Pgm phosphoglucomutase, Fbp fructose-1,6-bisphosphatase I, Glpx fructose-1,6-bisphosphatase II/sedoheptulose-1,7-bisphosphatase, Galu UTP-glucose-1-phosphate uridylyltransferase. Source data are provided as a Source Data file.

independent of heterologous transporters remained to be disclosed. In the glucokinase deficient strain SZ17, the generated glucose could be exported out of the cells through the heterologous glucose transporter GalP, avoiding over-accumulation of intracellular glucose. In comparison, regarding the SZ3 strain, genomic mutations would be required to facilitate glucose secretion, which could also be an explanation for the long-term passages required for the isolation of the homozygotes. To this end, whole-genome sequencing was performed to identify the functional mutations facilitating glucose secretion in SZ3. Based on published genome information of PCC 7942 (FACHB-805), types of mutations were characterized on the genome of SZ3 and SZ17 (Supplementary Table 1), most of which were generated during the preservation and passages of the PCC 7942 strain in our laboratory and were previously identified through genome sequencing assays. Through the comparisons between SZ3 and SZ17, a specific SNP (G274A) was identified on the *synpcc7942_1161* gene of the SZ3 genome, on which the 92nd valine was converted into isoleucine (Fig. 5a). The knockout of *synpcc7942_1161* (G274A) in the SZ3 strain (SZ141, Supplementary Fig. 9) reduced the glucose secretion by 78% to 0.33 g/L, thereby demonstrating the essential role of this gene (Fig. 5b).

*Synpcc7942_1161* was annotated to encode a divalent metal transporter with five transmembrane domains (Supplementary Fig. 10). Therefore, we hypothesized that the G274A mutation might improve the affinities of glucose to the *Synpcc7942_1161* inner membrane exporter, thereby facilitating more efficient glucose secretion in the SZ3 strain. However, a subsequent transcriptome assay suggested more complex effects of the *synpcc7942_1161*-G274A mutation. And among the transcripts with significantly elevated abundances in the SZ3 strain, the *synpcc7942_1161* transcript was the most significantly up-regulated, reaching up to a 77-fold increase. In addition, four other genes located in the same operon with *synpcc7942_1161* were also significantly up-regulated (*synpcc7942_1160* by 37 folds, *synpcc7942_1162* by 59 folds, *synpcc7942_1163* by 37 folds, and *synpcc7942_RS13740* by 34 folds), constituting the top five up-regulated transcripts in SZ3. To explore if the elevated transcriptions were caused by the *synpcc7942_1161*-G274A mutation, we introduced this mutation into the PCC 7942 WT genome (Fig. 5c and Supplementary Fig. 11) and examined the transcriptome changes. The results revealed that the *synpcc7942_1161*-G274A mutation significantly increased the transcript abundance of three gene-encoding potential transporter proteins (Fig. 5c and Supplementary Data 1), which

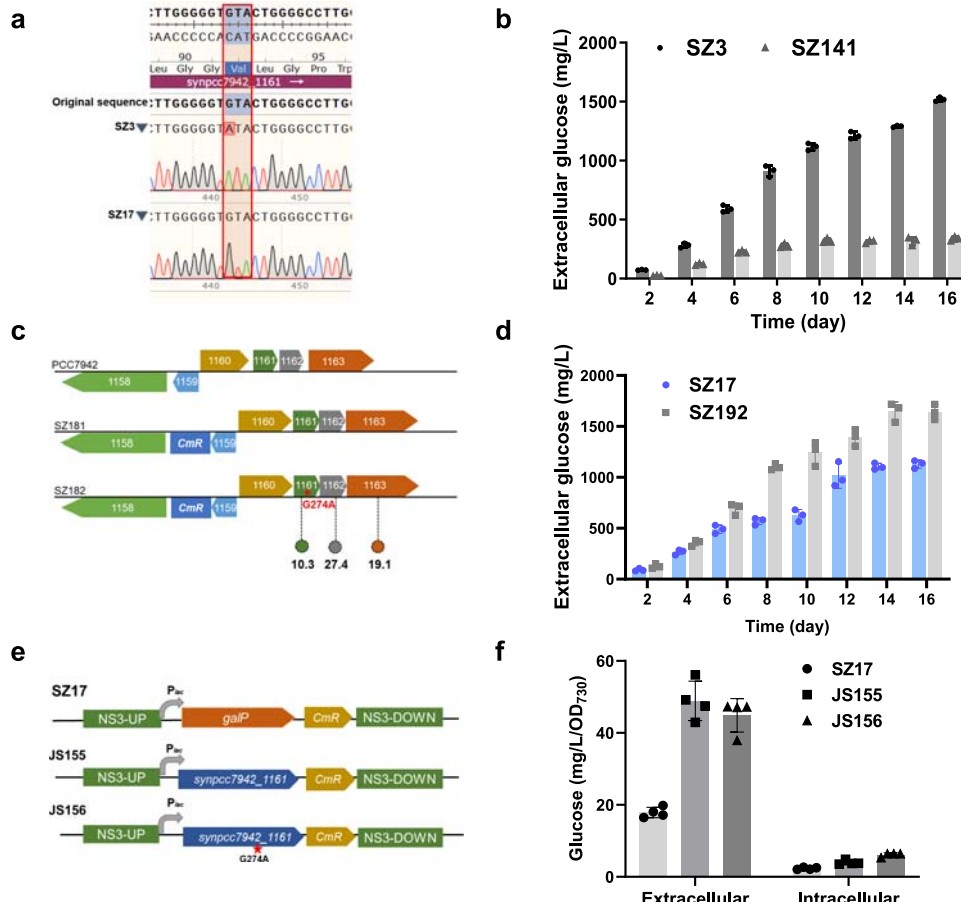

**Fig. 5 | Effects of** *synpcc7942_1161*- **G274A mutation on glucose secretion of PCC 7942 strains. a** Sanger DNA sequencing of the amplified *synpcc7942_1161* gene region of SZ3 and SZ17. **b** Effect of *synpcc7942_1161* gene deficiency on glucose secretion of SZ3 strain. **c** The construction strategy of the SZ182 strain by introducing the *synpcc7942_1161*- G274A mutation into the genome of PCC 7942; SZ181 carrying the same inserted antibiotics resistance marker was used as a control strain. The numbers indicate the elevated folds of respective transcript abundances in SZ182 compared to SZ181. **d** Overexpression effect of *synpcc7942_1161* (SZ192) on

glucose production of the SZ17 strain. **e** The construction strategy to replace the *galP* gene in SZ17 with *synpcc7942_1161* (with or without G274A mutation). **f** Comparisons of the effects of *synpcc7942_1161* and *synpcc7942_1161-G274A* on promoting glucose secretion when used for replacing GalP on the chromosome of SZ17. Data are presented as mean values ± SD. All sample size *n* (biological replicates) in (**b**, **d** and **f**) is 3. All sample size n (biological replicates) in (**c**) is 6. Source data are provided as a Source Data file.

suggested that the enhanced expressions of the operon were likely caused by this mutation.

*Synpcc7942_1161* abundance seemed to determine the glucose secretion phenotypes because the overexpression of the wild-type gene in the SZ17 strain (SZ192, Fig. 5d and Supplementary Fig. 12) increased the glucose production to the level of SZ3. We further replaced the GalP on the chromosome of SZ17 with wild-type *synpcc7942_1161* and a mutated version *synpcc7942_1161*-G274A, obtaining JS155, and JS156, respectively (Fig. 5e). As shown in Fig. 5f, JS155 and JS156 secreted comparable amounts of glucose into the culture broth, both of which were higher than SZ17. And the intracellular glucose concentrations in the two strains were also slightly improved. Thus, it can also be proposed that the *synpcc7942_1161*-G274A mutation promoted glucose secretion (in SZ3) mainly by elevating transcript abundances of the *synpcc7942_1161* rather than optimizing the specific activities of the respective transporter. Besides, we further evaluated and compared the stability of *synpcc7942_1161* mRNA in JS155 and JS156 under the control of the *Plac* promoter, and no difference was found between the two versions of *synpcc7942_1161* mRNA with or without the G274A mutation (Supplementary Fig. 13). Thus, it could be speculated that the elevated abundances of *synpcc7942_1161* transcripts were due to the increased transcription activities rather than the optimization of transcript stability. We also

assumed that the mutated *synpcc7942_1161* might contribute to a unidirectional transport function of intracellular glucose because the PCC 7942 WT strain carrying the *synpcc7942_1161*-G274A mutation could not absorb glucose (Supplementary Fig. 14). Based on the above results, we assumed a model for the generation of the glucokinase-deficient strain SZ3. As shown in Fig. 6, the glucokinase deficiency blocked the glucose-6-phosphate ↔ glucose cycle, and it enriched the spontaneous mutation of *synpcc7942_1161*-G274A. The *synpcc7942_1161*-G274A might promote glucose secretion by increasing the abundance or activity of the *Synpcc7942_1161* inner membrane exporter, which initiated the glucose secretions more efficiently. Finally, the elevated glucose secretions buffered the potential feedback stress and enabled the isolation of the final homozygote of SZ3.

### Engineering for enhanced and prolonged glucose production

The global RNA-sequencing analysis revealed that transcriptome was remodeled in SZ3, and expressions of 581 genes were significantly regulated (227 up-regulated and 354 down-regulated, Supplementary Data 2) compared with WT. Enrichment analysis revealed that the pathways involved in photosynthesis and oxidative phosphorylation were significantly regulated ($p$ value < 0.01) in SZ3 (Supplementary Fig. 15), and that was consistency with the weakened photosynthesis activities (oxygen evolution rates; Supplementary Fig. 16) in SZ3

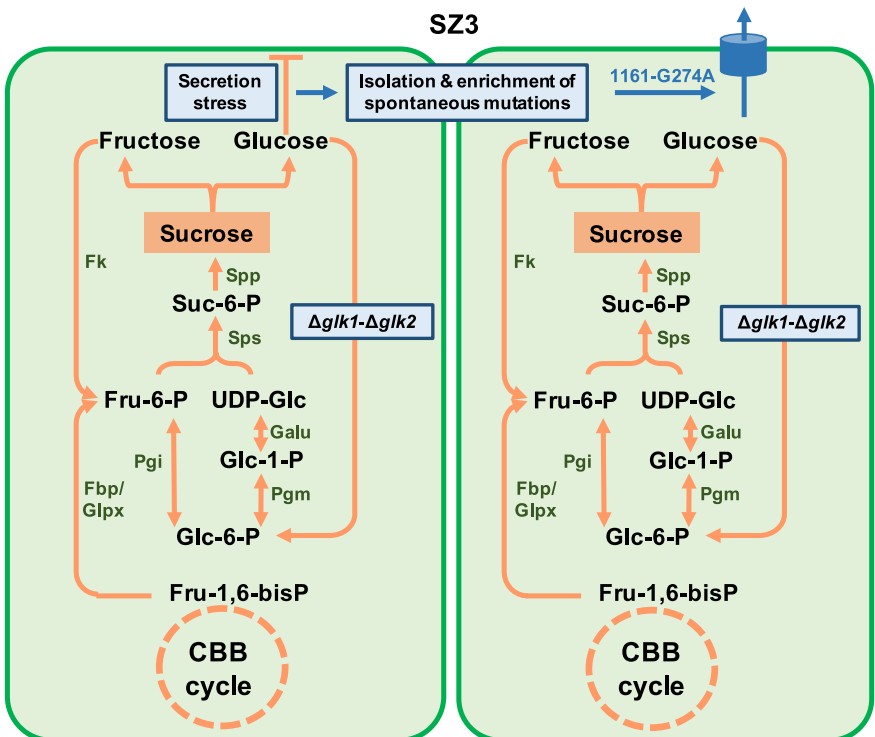

**Fig. 6 | Schematic diagram of genome mutation isolating mechanism promoted by glucokinase deficiency.** CBB cycle Calvin–Benson–Bassham cycle, Fru-1,6-bisP fructose-1,6-bisphosphate, Fru-6-P fructose-6-phosphate, UDP-Glc UDP-glucose, Suc-6-P sucrose-6-phosphate, Glc-6-P glucose-6-phosphate, Glc-1-P glucose-1-phosphate, Sps sucrose-phosphate synthase, Spp sucrose-phosphate phosphatase, Fk fructokinase, 1161: *synpcc7942_1161* gene, Pgi glucose-6-phosphate isomerase, Pgm phosphoglucomutase, Fbp fructose-1,6-bisphosphate I, Glpx fructose-1,6-bisphosphatase II/sedoheptulose-1,7-bisphosphatase, Galu UTP-glucose-1-phosphate uridylyltransferase.

compared with WT. Meanwhile, many genes participating in nitrogen metabolism and amino acids metabolism in SZ3 also show changed transcription levels, which were associated with the active "carbon spilling" status in the glucokinase deficiency strain, especially for those tightly associated with nitrogen metabolism. To further elucidate the effects of glucokinase deficiency metabolism, untargeted metabolomics assays were performed between SZ3 and the wild-type. Through LC-MS/MS assays, 116 differentiated metabolites were identified (Supplementary Data 3, 4). Consistent with the transcriptome change, several amino acids concentration inside SZ3 were changed. For example, concentrations of arginine, glutamate, and glutamine, which are involved in the ornithine-ammonia cycle[33], were all reduced by over 50% (Fig. 7). Another representative metabolomics characteristic of the glucokinase-deficient strain SZ3 was the increased abundance of sugar compounds and decreased content of phosphorylated metabolites (Supplementary Tables 2 and 3). This phenomenon indicated that glucokinases might function as non-specific phosphorylases with broader substrates in the PCC 7942 metabolism, which could also explain the difficulty to isolate the glucokinase-deficient homozygotes. To confirm this hypothesis, we purified Glk1 and Glk2 of PCC 7942 and evaluated the activities of the two enzymes on phosphorylating sugars other than glucose. As shown in Supplementary Fig. 17, both the two glucokinases showed phosphorylation activities on some other sugars besides glucose. The unspecific activities of Glk2 toward the eight sugars all exceed 20% (of the specific activity for glucose), reaching up to 50% for fructose, mannose, and galactose. The in vitro assay results provided further evidence for the hypothesis that glucokinases may play an important role in phosphorylating and recycling sugars in PCC 7942. In addition, several phosphorylated metabolites with decreased concentrations (ribulose-5-phosphate, dihydroxyacetone phosphate, fructose-6-phosphate, and erythrose-4-phosphate) are involved in the CBB cycle (Fig. 7), for

which the decreased abundances could also be a result of the carbon flow deprivation caused by the glucose synthesis & secretion.

Regarding the sucrose-metabolism pathway directly supporting the glucose synthesis, the abundance of the important intermediate metabolites was differently regulated, indicating an unbalanced catalytic route (Fig. 7). Compared to the wild-type control, the intracellular sucrose concentration decreased by 74.3%, whereas that of fructose increased by eightfold. These results indicate a strong driving force for glucose secretion and enhanced fructose output, which exceeded the conversion capacity of native fructokinase. The concentrations of most upstream metabolites generally decreased (by 88.3% for glucose-1-phosphate, 88.4% for glucose-6-phosphate, 69.9% for fructose-1,6-bisphosphate, and 97.7% for fructose-6-phosphate), except for UDP-glucose (2.93-fold increase). The functional enzymes responsible for the conversion of glucose-1-phosphate into UDP-glucose have not been identified in PCC 7942[34]. Therefore, the mechanisms causing the UDP-glucose over-accumulation should be further investigated in future studies.

To further improve the glucose photosynthetic production, several essential enzymes of the sucrose metabolism pathways were individually overexpressed. As shown in Fig. 8, the bifunctional sucrose-6-phosphate synthase/sucrose-6-phosphotase (catalyzing the synthesis of sucrose-6-phosphate and sucrose from fructose-6-phosphate and UDP-glucose), the invertase (catalyzing the degradation of sucrose into fructose and glucose), the phosphoglucose isomerase (catalyzing the conversion between fructose-6-phosphate and glucose-6-phosphate), the phosphoglucose mutase (catalyzing the conversion between glucose-1-phosphate and glucose-6-phosphate), the fructose-1,6-bisphosphatase (catalyzing the conversion from fructose-1,6-bisphosphate to fructose-6-phosphate) were all individually overexpressed in the SZ3 strain. Most recombinant strains performed enhanced glucose production, improving extracellular

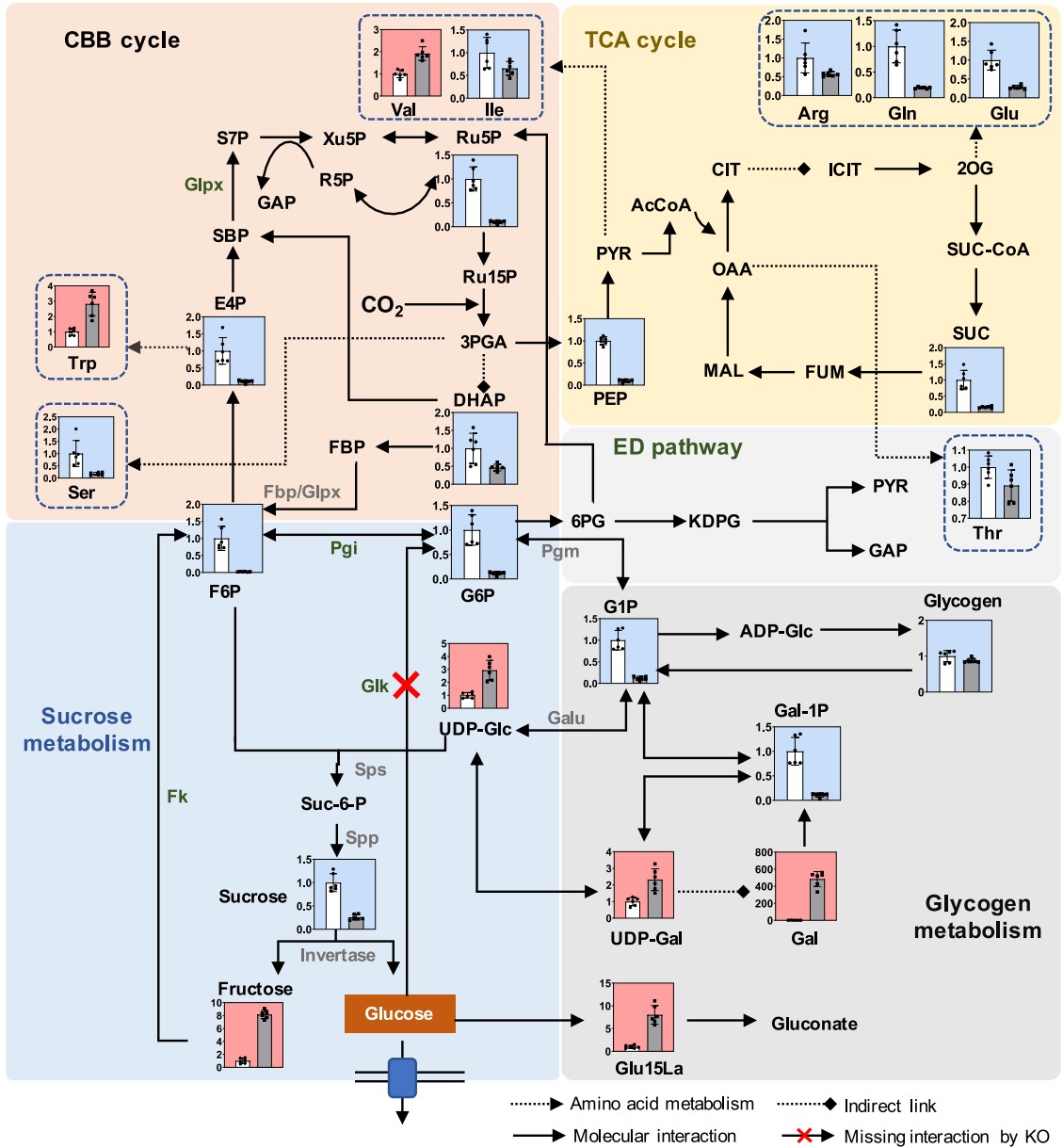

**Fig. 7 | Carbon metabolism in glucokinase-deficient SZ3 strain.** Relative amounts of intracellular metabolites of WT (white column) and SZ3 (gray column) grown in standard cultivation conditions (BG11 medium, 100 µmol photons/m²/s, air bubbling for carbon supply) for 8 d. Metabolites of SZ3 that significantly increased or decreased compared to WT are labeled red and blue, respectively. Data are presented as mean values ± SD. All sample size *n* (biological replicates) is 6. CBB cycle Calvin–Benson–Bassham cycle, TCA cycle tricarboxylic acid cycle, ED pathway Entner–Doudoroff pathway, FBP fructose-1,6-bisphosphate, Fbp fructose-1,6-bisphosphate I, Glpx fructose-1,6-bisphosphatase II/sedoheptulose-1,7-bisphosphatase, Galu UTP-glucose-1-phosphate uridylyltransferase, F6P fructose-6-phosphate, UDP-Glc UDP-glucose, Suc-6-P sucrose-6-phosphate, G6P glucose-6-phosphate, G1P glucose-1-phosphate, Sps sucrose-phosphate synthase, Spp sucrose-phosphate phosphatase, Fk fructokinase, Glk glucokinase, E4P erythrose-4-phosphate, SBP sedoheptulose-1,7-bisphosphate, S7P sedoheptulose-7-phosphate, GAP glyceraldehyde-3-phosphate, R5P ribose 5-phosphate, Xu5P xylulose-5-phosphate, Ru5P ribulose-5-P, Ru15P ribulose-1,5-diphosphate, DHAP dihydroxyacetone phosphate, ADP-Glc adenosine diphosphate glucose, Pgm phosphoglucomutase, Pgi glucose-6-phosphate isomerase, 3PGA 3-phosphate-glycerate, 6PG 6-Phosphogluconic acid, KDPG 2-keto-3-deoxy-6-phosphogluconate, PYR pyruvate, Glu15La Glucono-1,5-lactone, UDP-Gal UDP-Galactose, Gal Galactose, Gal-1P Galactose 1-phosphate, PEP phosphoenolpyruvate, AcCoA acetyl-CoA, CIT citrate, ICIT isocitrate, 2OG 2-oxoglutarate, SUC-CoA succinyl CoA, SUC succinate, FUM Fumarate, OAA Oxaloacetate, MAL Malate. Source data are provided as a Source Data file.

glucose concentrations from approximately 1.6 to 2 g/L during a 16-18 days cultivation process. The only exception was the overexpression of fructokinase, for which glucose secretion was decreased. A possible explanation is that the increased fructokinase enzymes catalyzed glucose as a non-specific substrate and restored the previously blocked glucose/phosphoglucose cycle (Supplementary Fig. 18), leading to the reduction of the glucose synthesis branch.

As mentioned above, the optimization of carbon supply and illumination strengths improved the glucose production of SZ3 and SZ17 strains, as well as the biomass accumulation rates. We used the SZ123 strain to improve glucose production by cultivation optimization, and the results showed improved biomass and glucose production (from 2 to 2.9 g/L; Fig. 9a; Supplementary Fig. 19) using concentrated (2×) BG11 culture medium[35]. For prolonged glucose synthesis, we adopted a fed-batch strategy by re-suspending the glucose-synthesizing cells (meaning harvesting the cells and suspending the cells with fresh medium) for every 12 d. This strategy effectively prolonged the glucose-synthesizing activities of the SZ123

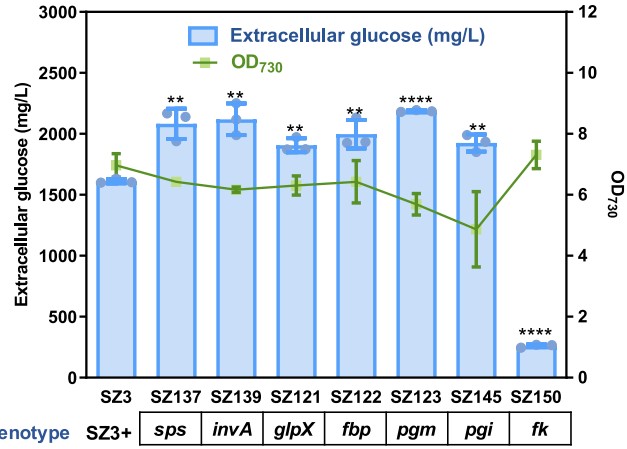

**Fig. 8 | Effects of regulating sucrose metabolism-related pathways on the SZ3 strain, and optimization of cultivation conditions for the SZ123 strain.** Extracellular glucose production and cell density of mutant strains calculated after a 15–18 d cultivation. The results for SZ145 and SZ150 are the extracellular glucose production from the 16th and 15th day of culture, respectively. Statistical analysis was performed using two-tailed unpaired Student's $t$ test (**$p < 0.01$, ****$p < 0.0001$). The $p$ values are (from left to right) 0.0029, 0.0026, 0.0011, 0.0049, <0.0001, 0.0017, <0.0001. Data are presented as mean values ± SD. All sample size $n$ (biological replicates) is 3. Source data are provided as a Source Data file.

cells (Fig. 9b) and led to the accumulation of 5 g/L glucose after 3 batches, which took up over 70% of the carbon flux fixed by the recombinant *Synechococcus* cells (Supplementary Fig. 20). The glucose accumulation rates gradually decreased with the decline in cell density. Therefore, the cellular robustness of the recombinant cell factories to maintain stable growth and metabolism in long-term cultivation is the main factor for the control of glucose production. Accordingly, future systematic solutions to strain and cultivation engineering should further improve these aspects.

## Discussion

In this work, we unlocked the potential of cyanobacterial photosynthesis to convert carbon dioxide directly into glucose. Photosynthesis is the most extensive biochemical process on Earth, and photoautotrophs convert solar energy and carbon dioxide into primary organic carbon to support the maintenance of the biosphere[36]. Currently, the refining of sugar-rich plant/algal biomass serves as the dominating route to produce sugar in large quantities, meeting the needs of the food and medicine areas as well as the biorefinery industry[8,9]. The "one-pot, one-step" mode of cyanobacterial photosynthetic production could serve as a more continual, stable, and efficient sugar alternative than the "plants-biomass-sugar" route[37]. Most cyanobacteria species can naturally synthesize and accumulate sugar-type compatible solutes to resist abiotic environmental stress[38]. The accumulation and secretion of these compatible sugar compounds (such as sucrose and trehalose) by cyanobacteria can be optimized through genetic manipulation[20,21], and biotechnology application scenarios such as the development of artificial consortia are explored and expanded[39,40]. However, glucose, the most representative and important monosaccharide molecule, cannot be efficiently synthesized and accumulated in cyanobacteria. Recently, glucose secretion (accompanied by sucrose secretion) has been observed in a PCC 7942 mutant with deleted *xpk* gene (which encodes phosphoketolase). However, this glucose synthesis can only be triggered in the dark with salts stress conditions, and the titer is rather limited, reaching approximately 3 mM by high-density cells (the initial OD$_{730}$ is 20, and the final average productivity of 0.027 g/L/OD$_{730}$)[24]. In

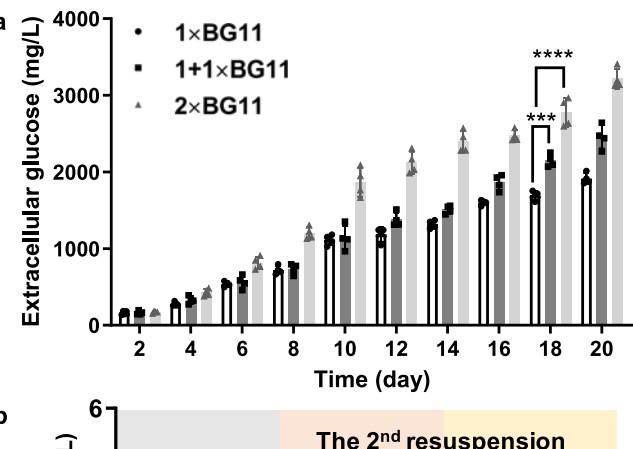

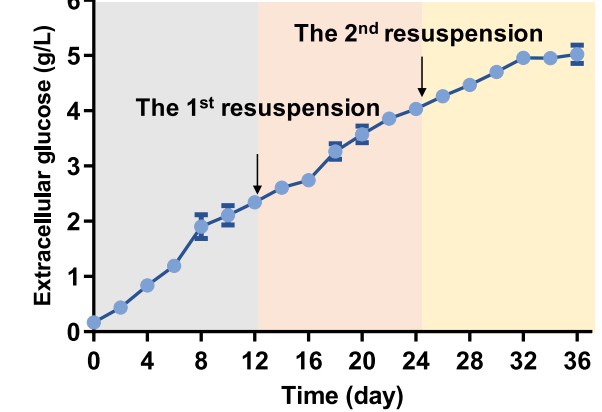

**Fig. 9 | Optimization of cultivation conditions of SZ123 strain. a** Effect of culture media supplementation on glucose production of SZ123 strain. Statistical analysis was performed using two-tailed unpaired Student's $t$ test (***$p < 0.001$, ****$p < 0.0001$). The $p$ values are (from left to right) 0.0001, <0.0001. **b** Resuspension-continuous culture of SZ123 strain based on (2×) BG11 medium. Data are presented as mean values ± SD. All sample size $n$ (biological replicates) in (**a**) is 4. All sample size n (biological replicates) in (**b**) is 3. Source data are provided as a Source Data file.

another, earlier reported work, the titer of salt-induced glucose synthesis (accompanied by fructose secretion) reached only 30 μM (productivity of 0.027 g/L/OD$_{750}$) in PCC 7942 by overexpressing sucrose hydrolase and hexose transporters[23]. In contrast, continuous synthesis of glucose during the rapid photoautotrophic growth phase was achieved in this work, with productivity higher than 0.27 g/L/OD$_{730}$, by simply removing the glucokinase activities in *Synechococcus* cells, meaning tenfold higher photosynthetic productivities in the absence of any heterologous catalytic enzymes or transporters and independent of any environmental stress inductions.

Although a majority of the isolated cyanobacteria species perform autotrophic metabolism and are deficient to absorb and utilize exogenous glucose as the carbon source, some cyanobacteria strains are capable of using glucose[41,42]. As for the cyanobacteria strains able to perform heterotrophic or mixotrophic metabolism using glucose, glucokinase plays an essential role in phosphorylating glucose and initiating the diverse glycolytic processes[43,44]. However, glucokinase genes are ubiquitous in the vast majority of the sequenced cyanobacterial genomes (Supplementary Data 5). For example, *Synechococcus elongatus* PCC 7942, which has been shown to lack the capacity to absorb and utilize exogenous glucose, has two glucokinase genes on its genome, implying that glucokinase has more physiological functions than glucose utilization. Through systematic genetic modifications, we identified an active and stable glucose-phosphoglucose cycle in PCC 7942, which was maintained in the sucrose metabolism pathway. Previously, sucrose synthesis and accumulation were generally recognized as a salt-stress responsive mechanism to resist

hyperosmotic stress[31,45]. In contrast, we demonstrated in this work that the synthesis-degradation cycle of sucrose in PCC 7942 remains active independent of salts stress induction, and the glucokinase activity performs as a "floodgate" to avoid the overaccumulation of glucose, which is an intermediate in the sucrose metabolism network and has a potential impact on the photosynthetic metabolism. Besides sucrose metabolism, some other non-specific reactions may also contribute to the synthesis and accumulation of glucose in PCC 7942. Thus, it could be assumed that intracellular glucose synthesis may be continuously maintained in PCC 7942, whereas glucokinase was crucial to recycle glucose into the central metabolism network by a phosphorylation process, and this mechanism can potentially rescue the potential energy loss and metabolism disturbance caused by glucose secretion and accumulation. In addition, although cell growths and photosynthesis were not inhibited (Supplementary Fig. 16), glucokinase-deficient strains showed improved sensitivity to 150 mM NaCl (Supplementary Fig. 21), indicating that the stable glucose-phosphoglucose cycle contributes to the rapid response of PCC 7942 toward salt stress. Previously it has been proven that the ionic effect is important for inducing sucrose accumulation in PCC 7942, whereby an elevated ion concentration directly activates the sucrose-synthesizing enzyme Sps and simultaneously inhibits the sucrose-degrading enzyme Inv[31]. The maintenance of the sucrose metabolism cycle permitted the direct activity-regulation of ion concentrations on the enzymes, and the glucokinase activities could guarantee the rapid re-utilization of sugars (fructose and glucose) from hydrolyzed sucrose (activity of Inv would be reduced rather than completely removed by elevated ion concentrations).

An important issue in microbial metabolic engineering is the adaptation of chassis cells to artificial pathways and genomic modifications. The in-plugged and rewired metabolic activities can disturb carbon distribution, co-factor supply, and redox balances of the native cellular metabolism, and are prone to stimulate the responses of the host cells on levels of physiology, metabolism, and even genetics[46], which should be relieved through metabolic designs and engineering. Transporter engineering has become a universal strategy to increase the adaptability of microbial chassis cells to heterologous pathways, which could promote product secretion, relieve the metabolic burden of intracellular over-accumulation and reduce the potential inhibitory effect of the products[47]. An alternative solution was suggested in this study. To achieve effective glucose transport, we introduced a heterologous transporter GalP on the chromosome of PCC 7942, which reduced the potential intracellular metabolic or physiological stress and facilitated the convenient and smooth acquisition of the genetic transformants (SZ17). In comparison, without introducing heterologous transporters, a spontaneous mutation (*Synpcc7942_1161*-G274A) was generated and enriched during the long-term cultivation process and endowed the mutant strain with even better capacities for glucose synthesis and secretion. Combining artificial genetic modifications with metabolic stress-induced spontaneous genomic mutations might rewire the metabolic flow more effectively.

In summary, we identified the key factors restricting the natural potential of cyanobacteria for the photosynthetic production of glucose. Without heterologous catalysis or transportation genes, a large portion of photosynthetically fixed carbon dioxide in the engineered strain with deficient glucokinase activities and a spontaneous genome mutation was rewired into the secretory production of glucose. The discoveries shed light on developing and industrializing more directional and continuous glucose production systems with solar energy and carbon dioxide.

## Methods

### Plasmid construction

All endogenous genes were PCR amplified from the wild-type PCC 7942 genome using gene-specific primers and high-fidelity DNA polymerase (TransGen, Beijing, China). All primers used in this study and the constructed plasmids are listed in Supplementary Data 6. The *E. coli* strain DH5α was used to construct the plasmids, and all strains were grown at 37 °C in the Luria-Bertani medium (liquid or agar petri dish) supplemented with corresponding antibiotics (Solarbio). Ampicillin, kanamycin, spectinomycin, gentamycin, and chloramphenicol were used at 100, 50, 50, 40, and 50 μg/mL, respectively, for strains with corresponding resistance genes. All backbone vectors were defined based on pUC19 or pBR322 by inserting homologous regions, promoters, and antibiotic resistance cassettes with restriction enzyme digestion, and ligation or seamless cloning. The homologous-recombination regions in all plasmids were located at approximately 1000 bp upstream sequence and 1000 bp downstream sequence of the integrated gene locus. *Eco*RI, *Xba*I, *Eco*RV, *Nae*I, *Eag*I, and *Bam*HI were the restriction cloning sites used to construct pSS1 and pSS2. All restriction enzymes were fast digestion enzymes from Thermo Fisher Science (Shanghai, China). The plasmids were constructed using the seamless cloning method and the Seamless Assembly Cloning Kit (Taihe Biology Co., LTD, China) according to the manufacturer's instructions. *E. coli* colonies with positive plasmids were confirmed by colony PCR using gene-specific primers and Taq DNA polymerase (TransGen Biotech). Plasmids purified with a Mini-prep kit (OMEGA Bio-Tek) were checked by Sanger sequencing before the PCC 7942 transformation.

### Construction and cultivation of PCC 7942 derived strains

The genetic modifications of the PCC 7942 genome were performed using a double homologous recombination system to integrate the genes into the target sites. All plasmids were transformed into PCC 7942 through the natural transformation. Transformants grown on antibiotic selective plates were analyzed by PCR and second-generation DNA sequencing of the amplified fragments[32]. The PCC 7942 derived strains constructed in this study are listed in Supplementary Data 7.

PCC 7942 derived strains were inoculated into liquid BG11 medium containing corresponding antibiotics and cultivated for 4–6 d under rotary shaking (150 rpm) and white-light illumination of 50 μmol photons/m²/s at 30 °C. Subsequently, the culture broth was inoculated into 200 mL of BG11 medium in 250 mL flasks under continuous white light of 100 μmol photons/m²/s and bubbled with air. Kanamycin, spectinomycin, gentamicin, and chloramphenicol were added into the medium for final concentrations of 20, 20, 2, and 10 μg/mL, respectively. Finally, the culture broth from the flasks was centrifuged and resuspended with 65 mL fresh BG11 culture medium to the initial $OD_{730}$ of approximately 2.0 in column photobioreactors (with a diameter of 3 cm). Except when specifically noted, the cultivations for photosynthetic glucose production were performed at 30 °C. During the long-term cultivation experiment, antibiotics were eliminated, 8 mM TES-NaOH (pH = 8) was added to maintain the pH of the culture, and the genotypes of the cells were checked by PCR at the end of the cultivation. During the cultivation process, 0.5 mL of culture broth was sampled from each column photobioreactor every other day to measure the $OD_{730}$ and glucose concentrations, and sterile water was supplemented to maintain the original volume. For special experiments, the organic carbon contents (citric acid and ferric ammonium citrate) would be removed from the BG11 medium.

### Glucose production

To determine the extracellular glucose contents secreted by the PCC 7942 derived strains, the culture broth was sampled and centrifuged at 12,000 ×*g* for 1 min. Then the glucose concentrations in the supernatant were calculated using the D-glucose assay kit (Megazyme) or ICS5000+ (DIONEX, Thermo Scientific, USA) ion-exchange chromatography system equipped with an electrochemical detector and a Dionex CarboPac MA1 analytical column (4×250 mm, Thermo

Scientific, Waltham, MA, USA). Intracellular glucose was extracted from cellular pellets following a previous study[48]. In brief, the cell pellets were resuspended in 1 mL 80% ethanol (volume to volume) and then incubated at 65 °C for 4 h. After centrifugation at 12,000 ×g for 5 min, the supernatant was transferred to a clean tube and dried under a stream of $N_2$ at 55 °C. Subsequently, the dry residues were dissolved in ultrapure water, and the dissolved solution was filtered into clean vials with a 0.22 μm filtration membrane. Glucose contents in the filtrate were also calculated using the D-glucose assay kit (Megazyme) or ion chromatography.

### Glucokinase activity

Glucokinase activity was measured by a modified enzyme-linked assay[3]. *Synechococcus* cells cultivated in BG11 medium were centrifuged and resuspended in 1 mL of breakage buffer (50 mM Tris-HCl, pH 7.4) pre-cooled to 4 °C. Quartz sand (Sigma) was added to the cell suspension, and the cells were disrupted by vortex mixing at 4 °C for 30 min. After centrifugation, 60 μL of the supernatant was transferred to a 96-well plate, to which 200 μL of a reaction buffer (100 mM Tris-HCl, pH 7.8, 2.5 mM ATP, 4 mM $MgCl_2$, 20 mM KCl, 0.2 mM $NADP^+$, 10 mM glucose, and 5 U/mL glucose-6-phosphate dehydrogenase) was added, and the absorbance at 340 nm was measured every 5 min in the dark. The slope of $A_{510}$, which increased with time, was calculated, and the specific enzyme activity of glucokinase in the reaction mixture was calculated according to the Lambert-Beer law.

### The ¹³C labeling cultivation

The $^{13}C$ labeling experiments were performed using $^{13}C$ labeled sodium bicarbonate (Cambridge Isotope Laboratories, 99% $^{13}C$, CLM-441), and $^{12}C$-$NaHCO_3$ was added as a control. First, cells at the exponential growth phase were adjusted to an $OD_{730}$ of 0.6 in 10 mL BG11 including 50 mM $NaHCO_3$ and appropriate antibiotics, and then cultivated in 30 mL sealed flasks (with no headspace) with or without citric acid and ammonium ferric citrate under continuous light conditions. For assays of the intracellular metabolites, cells were collected by centrifugation every 24 h, extracted rapidly with 2 mL of 80:20 methanol/water (vol/vol), and then frozen in liquid nitrogen. Intracellular metabolites were extracted via the freeze/thaw cycle five times. After centrifugation and nitrogen blowing, 100 μL $ddH_2O$ was added to the sample for resuspension and was taken for derivatization reaction. The sample derivatization was carried out in two steps: (1) adding 40 μL methoxyamine solution (25 mg/mL, dissolved in pyridine solvent) and then incubating at 60 °C for 30 min. (2) 40 μL TMS reagent (BSTFA/TMCS, 99:1) was added to each sample and then reacted at 37 °C for 120 min. For assays of the extracellular glucose, 100 μL supernatant of SZ3 culture broth was taken for freeze-drying and derivatization respectively. After centrifugation, GC-MS was performed to analyze the samples. GC-MS assays were carried out by Agilent 7890, equipped with Agilent 5977 mass detector and HP-INNOWax column (30 m × 0.32 mm × 0.25 μm). Ultra-pure helium was used as carrier gas at a constant flow rate of 3 mL/min. The temperature of the column box was initially kept at 60 °C for 5 min, increased to 260 °C at a rate of 10 °C /min, and then kept at 260 °C for 10 min. Characteristic mass signals from glucose were detected at $m/z$ 204(M + 0), 205(M + 1), 206(M + 2), and then were calculated based on the ratio of labeled metabolites to unlabeled, to analyze related intermediates and to detect the labeled ratio of glucose in cells per unit time.

### Genome sequencing

For whole-genome re-sequencing of SZ3 and SZ17, genomic DNA was isolated and analyzed for qualities and concentrations using the Nanodrop ND-1000 system (Thermo Scientific, US) and Qubit fluorometer (Thermo Scientific, US). Subsequently, quantified DNA samples were fragmented, blunted, modified with 3′-A overhangs, ligated to Illumina's standard sequencing adapters, and amplified by PCR. The library was sequenced as paired-end reads using an Illumina HiSeq 2500 sequencer. Library constructions and sequencing were performed by Allwegene Tech (Beijing, China). The reference sequence (*Synechococcus elongatus* PCC 7942, FACHB-805) was obtained from the GenBank for read mapping. BWA[49] was used to reference sequences, and SAMtools[50] was used to sort the results and mark duplicate reads. Subsequently, SNPs and structure variation (SV) between reference sequences and sample sequences were identified by SAMtools and BreakDancer[51], respectively. Then, DNA fragments containing SNPs and SV were amplified from the genomic DNA and further verified by Sanger sequencing.

### *Synpcc7942_1161* mRNA stability evaluation

*Synechococcus* cells cultivated in flasks shaken under 50 μmol photons/m²/s white fluorescent light at 30 °C. Cells in the log phase were harvested and resuspended to an $OD_{730}$ of 4. Actinomycin D (ActD, 5 μg/mL) was added to block de novo mRNA synthesis and total RNA was isolated at indicated time points (0, 5, 25, and 45 min) after adding ActD using a bacterial RNA extraction kit (Vazyme, Nanjing, China). The RNA samples were then reverse transcribed into cDNA using HiScript III RT SuperMix for qPCR (Vazyme, Nanjing, China). Quantitative RT-PCR was performed on a LightCycler 480 Sequence Detector (Roche, Basel, Switzerland; LightCycler 480 software 1.5) based on SYBR Green I fluorescence (ChamQ Universal SYBR qPCR Master Mix-Vazyme, Nanjing, China).

### Untargeted metabolomics analysis

*Synechococcus* cells cultured to the 8th day under standard cultivation conditions (30 °C, 100 μmol photons/m²/s, air bubbling for carbon supply) were centrifuged and washed three times with a phosphate buffered saline buffer (Supplementary Fig. 22). The cell pellets were frozen in liquid nitrogen for 15 min and then stored at −80 °C until the analysis. Afterward, 200 μL of $ddH_2O$ and 800 μL of an acetonitrile-methanol mixture (1:1) were added to 80 mg of cell pellets. After vortexing for 1 min, all samples were stored at −20 °C for 1 h for protein precipitation. The samples were then centrifuged at 12,000 × g for 20 min at 4 °C. The supernatant was collected and evaporated to dryness in a vacuum concentrator. Subsequently, 100 μL of resuspension solution (0.1% formic acid in $ddH_2O$) was added and vortexed for 30 s. The resulting solution was filtered through a 0.22 μm membrane. To ensure the quality of the metabolomics analysis, quality control (QC) samples were prepared by mixing six replicates in each group.

For the untargeted metabolomics analysis, samples were injected into an Agilent 1290 Infinity Ultra-High Performance Liquid Chromatography equipped with an Acquity UPLC BEH Amide column (2.1 mm × 100 mm, 1.7 μm particle size) and detected with an AB Triple TOF 5600/6600 mass spectrometer equipped with an electrospray ionization (ESI) source (AB SCIEX, Canada). The temperature of the column was maintained at 25 °C. The gradient mobile phase was a mixture of 25 mM ammonium acetate and 25 mM ammonia in water for mobile phase A, and acetonitrile for mobile phase B, at a flow rate of 0.30 mL/min. The proportion of mobile phase B was optimized as 0–0.5 min, 95% B; 0.5–7 min, 95%–65% B; 7.0–8.0 min, 65%–40% B; 8.0–9.0 min, 40% B; 9.0–9.1 min, 40%–95% B; 9.1–12 min, 95% B.

A scan range from 50 to 1200 $m/z$ (mass-charge ratio) was adopted for the mass spectrometry (MS) analysis. The ion spray voltage was set to +5 kV for the positive ionization mode (ESI + ) and -5 kV for the negative mode (ESI-). The collision energy was set to +15 and −15 eV for ESI+ and ESI-, respectively. The declustering potentials were set to +60 V (ESI+) and −60V (ESI−), and the source temperature was 650 °C. The raw LC-MS data was converted by the ProteoWizard software and introduced into an XCMS for peak identification, matching, and integration. Compound identification of metabolites by MS/MS spectra with an in-house database established with available authentic standards. The identified metabolites were further searched against the

Kyoto Encyclopedia of Genes and Genomes database for the metabolic pathway analysis. Subsequently, data were pre-processed by Pareto-scaling for the principal component analysis, partial least-squares discriminant analysis (PLS-DA), and orthogonal partial least-squares discriminant analysis (OPLS-DA). The one-dimensional statistical analysis included the student's *t* test and multiple variation analysis.

## Transcriptome profiling

*Synechococcus* cells cultured to the 4th day under standard cultivation conditions (30 °C, 100 μmol photons/m²/s, air bubbling for carbon supply) were centrifuged and washed with DNase/RNase-free water (Supplementary Fig. 23). Cell pellets were collected and quick-frozen in liquid nitrogen for 15 min. TRIzol reagent (Invitrogen) was used for total RNA extraction from frozen cell pellets (three biological replicates in each group). The RNA quality and concentration were analyzed using the Nanodrop ND-1000 system (Thermo Scientific, USA) and Agilent 2100 RAN NANO 6000 Assay kit (Agilent Technologies, CA, USA). Paired-end libraries were prepared using the RNA Library Prep Kit (Illumina). The enriched mRNA was fragmented by adding a fragmentation buffer, and the first-strand cDNA was synthesized using random hexamer primers and reverse transcriptase, for which mRNA fragments were used as templates, followed by second-strand cDNA synthesis using DNA polymerase I, RNaseH, buffer, and dNTPs. The synthesized double-stranded cDNA fragments were then purified using a QIAQuick PCR kit. The purified double-stranded cDNA was poly-adenylated and adapter-ligated to prepare the paired-end library. The adaptor-ligated cDNA and adaptor primers were used for PCR amplification. Finally, the Illumina platform was applied to sequence the resulting cDNA libraries, and 150 bp paired-end reads were obtained. Clean reads were trimmed from raw reads, and all downstream analyses were based on the clean reads. By using the Bowtie 2 software, paired-end clean reads were aligned to the reference genome of *Synechococcus elongatus* PCC 7942. The number of reads corresponding to each gene was calculated using FeatureCounts[52]. Subsequently, every gene's fragments per kilobase per million (FPKM) were calculated based on the length of the gene and the reads count mapped to this gene. Genes with *p*-adj <0.05 and |log₂foldchange| > log₂(1.5) were defined as differentially expressed genes (DEGs).

## Protein expression and purification

The Glk1 (*Synpcc7942_0221*), Glk2 (*Synpcc7942_2111*), and Fk (*Synpcc7942_0116*) genes were amplified from the PCC 7942 genome and cloned into the pET28a, generating the recombinant expression plasmids pYW16, pJS89 and pJS90, respectively. The recombinant plasmids were transformed into *E. coli* strain BL21 (DE3) for the overproduction of target proteins. The transformants were grown in 500 mL LB medium containing 50 μg/mL kanamycin at 37 °C. At an OD₆₀₀ of 0.6–0.8, cells were induced by 0.3 mM IPTG at 16 °C for 20 h. After induction, cells were then harvested by centrifugation and disrupted by sonication in prechilled lysis buffer (20 mM Tris-HCl, pH 7.8). The lysates were centrifuged at 20,000 × *g* for 60 min, and the resulting supernatants were filtered through 0.22 μm polyethersulfone membranes and then loaded on a Ni Sepharose 6 FF column (GE Healthcare, Chicago, IL) for affinity purification. The typical bind-wash-elute procedure was then used to get the target proteins. Protein concentration in the fractions was determined by the Bradford method, and the purity of target proteins was examined by SDS-PAGE.

## Sugar phosphorylating activities assays

The phosphorylating activities of glucokinase and fructokinase on different sugar substrates were measured by using the pyruvate kinase/lactate dehydrogenase (PK/LDH) enzyme-linked assay as previously described with modifications[53]. In brief, 20 μL of the purified protein was transferred to a 96-well plate, to which 210 μL of a reaction buffer (100 mM Tris-HCl, pH 7.8, 2.5 mM ATP, 4 mM MgCl₂, 20 mM KCl, 0.3 mM NADP⁺, 10 mM specific sugar, 1 mM PEP, 1.4 U/mL PK, 2.8 U/mL LDH) was added, and the absorbance at 340 nm was measured every 15 s during 1 h in the dark. The slope of A₅₁₀, which decreased with time, was calculated, and the specific enzyme activity of glucokinase/fructokinase on each sugar would be compared to the activities on glucose/fructose to get the relative ratios.

## Glucokinase genes distributions analysis

To comprehensively investigate the genomic Glucokinase sequences of cyanobacteria, we downloaded all available cyanobacterial genomic sequences and their annotation from the NCBI assembly database using the NCBI-datasets tools. A total of 986 no-redundantly reference-level cyanobacteria genomes were used to construct the genome database. The genome database was screened using the hmmsearch tool of the Hmmer package (version 3.1b2)[54], and potential Glucokinase sequences from 947 cyanobacterial genomes were retrieved to show significant matches with this Glucokinase HMM (PF02685) with E values < 1⁻⁵⁰ (Supplementary Data 5).

## Photosynthetic oxygen evolution rates and respiration rates

*Synechococcus* cells were harvested and resuspended with 17 mL fresh BG11 to the initial OD₇₃₀ of approximately 2.0. The respiration rates were measured under dark for 200 s at 30 °C and the whole chain photosynthetic oxygen evolution rates were measured for 160 s at each light level (RGB light source) with the YZQ-201A photosynthetic instrument (Yizongqi Technology Co., Ltd, China) in the presence of 10 mM NaHCO₃. The total photosynthetic oxygen evolution rates were calculated as the net photosynthetic rate plus the respiration rate.

## Reporting summary

Further information on research design is available in the Nature Portfolio Reporting Summary linked to this article.

# Data availability

Whole genome sequencing and transcriptome sequencing datasets reported in this paper are available through NCBI BioProject accession PRJNA740138. All raw data for the untargeted metabolomics analysis have been deposited into CNGB Sequence Archive (CNSA)[55] of China National GeneBank DataBase (CNGBdb)[56] with accession number CNP0004406. The identified metabolites were searched against the Kyoto Encyclopedia of Genes and Genomes database [https://www.kegg.jp/kegg/kegg2.html]. Compound identification from MS/MS spectra was conducted with an in-house database generated from available authentic standards. The reference genome and gene model annotation files of *Synechococcus* elongatus PCC 7942 and FACHB-805 were obtained from GenBank [https://ftp.ncbi.nlm.nih.gov/genomes/all/GCA/000/012/525/GCA_000012525.1_ASM1252v1/]. Source data are provided with this paper.

# Code availability

Input data files used for all scripts, the generated output files and script for the analysis of the genomic Glucokinase sequences of cyanobacteria are available at GitHub [https://github.com/yudifeiluo/cyanobacteria][57].

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

## Acknowledgements

The research was supported by the National Key Research and Development Program of China (Grant number 2021YFA0909700, to X.L. and G.L.), the Youth Innovation Promotion Association CAS (to G.L.), the National Natural Science Foundation of China (Grant number 32070084 to G.L., 32270103 to G.L., 32271484 to X.L.), the DNL Cooperation Fund, CAS (DNL202014, to G.L.), and the Shandong Taishan Scholarship (to X.L. and G.L.).

## Author contributions

G.L and X.L designed the research; S.Z, J.S., D.F., X.Z., Y.W., H.S., J.C., and G.L. performed the research; S.Z., J.S., G.L., and X.L. analyzed the data; and S.Z., J.S., G.L., and X.L wrote the manuscript.

## Competing interests

The authors declare no competing interest.
