## [Peer Review File · Nature Communications]

Unlocking the potentials of cyanobacterial photosynthesis for directly converting carbon dioxide into glucoseEditorial Note: This manuscript has been previously reviewed at another journal that is not operating a transparent peer review scheme. This document only contains reviewer comments and rebuttal letters for versions considered at Nature Communications.

Reviewers' Comments:

Reviewer #1:

Remarks to the Author:

The authors have done an outstanding job in responding to reviewers comments. The revised manuscript is much improved and presents a more complete understanding of cyanobacterial sugar metabolism.

A general language editing can improve clarity of the text and figure legend.

Reviewer #2:

Remarks to the Author:

The manuscript of Zhang et al has been significantly revised. The paper provides novel insights into sugar metabolism in cyanobacteria. The revision provides important new data on specificity profiles of Glk, as well as a better connection between the transcriptomics and metabolomics data. These additional enzyme assays show that Glk may also phosphorylate fructose and other sugars.

It is my opinion that the authors have addressed reviewer concerns. I would recommend a careful read-through of the new text to catch grammar mistakes.

Reviewer #3:

The author has addressed my concerns except for one regarding Figure R10.

The figure legend mentions "204 (M + 0)," which, according to my understanding, represents the absence of carbon coming from $\text{NaH}^{13}\text{CO}_3$ in glucose. However, I am unclear as to what exactly 204 represents. Is it the mass of glucose, or does it represent something else?

Point-to-point replies to the reviews of manuscript NCOMMS-23-05669A

REVIEWER COMMENTS

Reviewer #1 (Remarks to the Author)

The authors have done an outstanding job in responding to reviewers comments. The revised manuscript is much improved and presents a more complete understanding of cyanobacterial sugar metabolism.

A general language editing can improve clarity of the text and figure legend.

RESPONSE: Thanks for the comments. We would like to express our gratitude for the thoughtful comments and constructive suggestions from all the reviewers, and we have revised the language of the manuscript.

Reviewer #2 (Remarks to the Author)

The manuscript of Zhang et al has been significantly revised. The paper provides novel insights into sugar metabolism in cyanobacteria. The revision provides important new data on specificity profiles of Glk, as well as a better connection between the transcriptomics and metabolomics data. These additional enzyme assays show that Glk may also phosphorylate fructose and other sugars.

It is my opinion that the authors have addressed reviewer concerns. I would recommend a careful read-through of the new text to catch grammar mistakes.

RESPONSE: Thanks for the comprehensive and accurate summary of this work and for the positive comments. We have read through the manuscript and corrected the grammar mistakes.

Reviewer #3 (Remarks to the Author)

The author has addressed my concerns except for one regarding Figure R10.

The figure legend mentions "204 (M + 0)," which, according to my understanding, represents the absence of carbon coming from $\text{NaH}^{13}\text{CO}_3$ in glucose. However, I am unclear as to what exactly 204 represents. Is it the mass of glucose, or does it represent something else?

RESPONSE: Thanks for the suggestions. We have described the meaning of "204 (M + 0)" in "Methods" section, that the characteristic mass signals from glucose were detected at m/z 204(M+0), 205(M+1), 206(M+2), and then were calculated based on the ratio of labeled metabolites to unlabeled, to analyze related intermediates and to detect the labeled ratio of glucose in cells per unit time. The structure after glucose derivatization is shown in the following Figure R1.

Glucose, STMS derivative - MS Interpreter

File Edit View Options Help

Formula Calculator

m = 204 | C₂₁H₅₂O₆Si₅

Calculate Options Parent = 540 Loss = 336

m/z	mass	formula	loss	type	H	rate	abund
204	204.100183	C ₈ H ₂₀ O ₂ Si ₂	C ₁₃ H ₃₂ O ₄ Si ₃	dissociation	-1	-31	999

7I Ions	D+E	RDB	Mass	C	H	I
O ₄ Si ₅	Odd	6	203.86429	0	0	4
CO ₆ Si ₄	Odd	6	203.88228	1	0	4
CH ₄ O ₃ Si ₅	Odd	5	203.90068	1	4	4
C ₂ O ₆ Si ₃	Odd	6	203.90027	2	0	4
C ₂ H ₄ O ₄ Si ₄	Odd	5	203.91866	2	4	4
C ₂ H ₈ O ₂ Si ₅	Odd	4	203.93706	2	8	4
C ₃ H ₄ O ₅ Si ₃	Odd	5	203.93665	3	4	4
C ₃ H ₈ O ₃ Si ₄	Odd	4	203.95505	3	8	4
C ₃ H ₁₂ O ₅ Si ₂	Odd	3	203.97345	3	12	4
C ₄ O ₅ Si ₂	Odd	10	203.87995	4	0	4
C ₄ H ₄ O ₆ Si ₂	Odd	5	203.95464	4	4	4

Figure R1. The structure after glucose derivatization